# Magmatic plumbing and dynamic evolution of the 2021 La Palma eruption

Carmen del Fresno [1] ✉, Simone Cesca [2], Andreas Klügel [3], Itahiza Domínguez Cerdeña [4], Eduardo A. Díaz-Suárez [4], Torsten Dahm [2,5], Laura García-Cañada [1], Stavros Meletlidis[4], Claus Milkereit[2], Carla Valenzuela-Malebrán [2,5], Rubén López-Díaz[1] & Carmen López[1]

The 2021 volcanic eruption at La Palma, Canary Islands, was the island's most voluminous historical eruption. Little is known about this volcano's feeding system. During the eruption, seismicity was distributed in two clusters at ~10-14 km and ~33-39 km depth, separated by an aseismic zone. This gap coincides with the location of weak seismic swarms in 2017-2021 and where petrological data have implied pre-eruptive magma storage. Here we use seismological methods to understand the seismic response to magma transfer, with 8,488 hypocentral relocations resolving small-scale seismogenic structures, and 156 moment tensors identifying stress heterogeneities and principal axes flips. Results suggest a long-lasting preparatory stage with the progressive destabilisation of an intermediate, mushy reservoir, and a co-eruptive stage with seismicity controlled by the drainage and interplay of two localised reservoirs. Our study provides new insights into the plumbing system that will improve the monitoring of future eruptions in the island.

The 2021 La Palma eruption started on September 19 and lasted more than 85 days[1-4], forming a new edifice on the western flank of Cumbre Vieja volcano. It was the longest historical eruption at La Palma[5] and the most voluminous, with extruded magma volume[1] exceeding 0.2 km³ and forcing the evacuation of ~7000 residents. Despite hosting the highest number of historical eruptions in the Canarian Archipelago[5], little was known of the Cumbre Vieja feeding system before the 2021 eruption. The only information was provided by petrological[6-9] and gravity studies[10]. Earthquakes accompanying the 2021 unrest reached unprecedented rates and magnitudes, they provide a unique dataset to map magma pockets and pathways improving our knowledge of the dimensions and structure of the plumbing system as well as its dynamics during the eruption.

The Canary Islands is an archipelago of volcanic origin at the passive NW margin of the African plate. La Palma, NW of the archipelago (Fig. 1), is the second youngest island, with an emerged surface of 706 km² and a height of 2430 m a.s.l. Its subaerial activity began 1.8 Myr[11], following the formation of a seamount and its uplift during a period of intense magmatism, during which time the island underwent alternating eruptive episodes and flank collapses[11]. The successive formations of the Garafía, Taburiente, Bejenado and Cumbre Vieja volcanoes show a North to South migration of volcanic activity[11-14]. During the latest stage of island construction (123 kyr), the volcanic activity was concentrated in the southern sector, forming the Cumbre Vieja volcano, with eruption vents aligned along a ~20 km long NS oriented rift[15,16]. All 8 historical eruptions of the island took place in this region (Fig. 1), with the most recent ones in 1949[5,8], 1971[5,9] and 2021[1-4].

Barometric studies of lavas and xenoliths from Cumbre Vieja[6-8] suggest pre-eruptive magma storage occurs in upper mantle reservoirs at ~15-26 km depth and possibly also down to 50 km[9]. During eruptive episodes, magma ascends forming dikes and sills, temporarily stagnating at an accumulation or underplating zone at 7-15 km depth; this is also the main horizon where evolved magmas are thought to form[6]. Long-term seismicity precursors before the 1949[17,18] and 2021[14,19]

[1]Instituto Geográfico Nacional (IGN), Madrid, Spain. [2]GFZ German Research Centre for Geosciences, Potsdam, Germany. [3]Department of Geosciences, University of Bremen, Bremen, Germany. [4]Instituto Geográfico Nacional (IGN), Santa Cruz de Tenerife, Spain. [5]Institute of Geosciences, University of Potsdam, Potsdam-Golm, Germany. ✉e-mail: cdelfresno@mitma.es

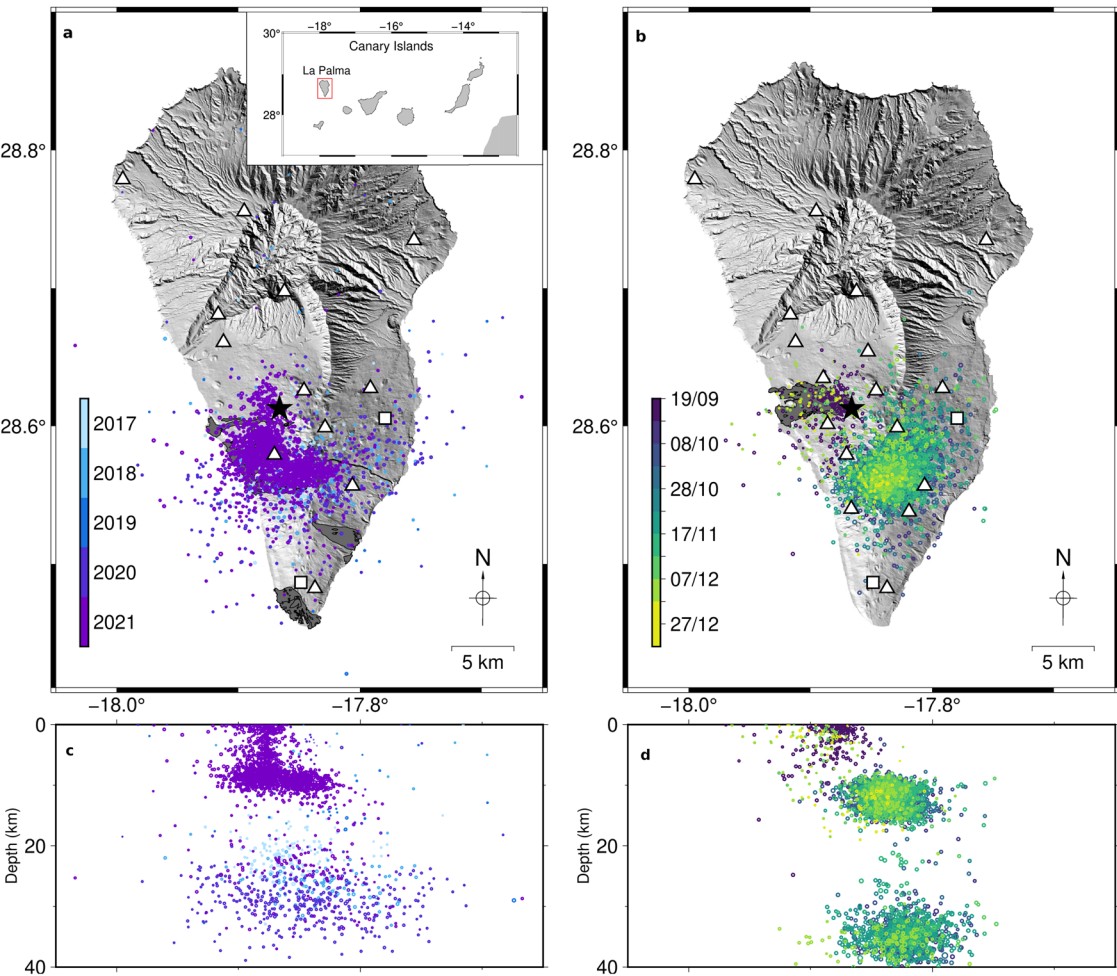

**Fig. 1 | Seismotectonic settings of La Palma Island, Canary Islands, Spain. a, b** Shaded-relief elevation model overlaid by the seismicity record at La Palma from the IGN catalogue (coloured dots) with seismic stations (triangles) and GNSS stations (squares) used in this work prior to (**a**) and after (**b**) eruption onset. Lava fields of previous historical eruptions (**a**) and this eruption (**b**) are shown in grey. Inset provides tectonic setting of the eruption within La Palma in the Canary Islands archipelago. **c, d** Longitude vs depth scatter plots of the data shown in panels **a** and **b**, respectively.

eruptions could indicate progressive magma accumulation in the pre-eruptive reservoirs months-to-years prior to eruptions.

The 2021 eruption is the first to be locally monitored at La Palma. In recent years, the Instituto Geográfico Nacional (IGN) seismic network[20] has been gradually improved, now counting 12 seismic stations on the island, plus 5 temporal stations installed during the eruption to support real-time procedures during the emergency period (Fig. 1). According to the IGN seismic catalogue, pre-eruptive anomalous activity beneath Cumbre Vieja began in October 2017[14,19], with a seismic swarm with more than 300 weak earthquakes (Local Magnitude, $M_L$[21] < 2.0) lasting ~10 days occurred at ~20 km depth. With the term swarm we refer to a series of earthquakes occurring closely in time and within a small volume, displaying a similar range of magnitude and not the usual mainshock-aftershocks pattern[22]. After some quiescence, a second, more populated swarm occurred in February 2018 at 25–30 km depth. The 2017 swarm was preceded by increases in hydrogen concentration and air-corrected helium isotopic ratio (R/Ra) c in the epicentral region, while the 2018 swarm was followed by increases of (R/Ra)c and thoron soil concentration, evidencing magmatic activity and unrest at depth[14,19]. Five more swarms occurred until June 2021, all characterised by $M_L \leq 1.9$, 20–35 km depths and 2–10 days durations. On September 11, 2021, a new series started in the region, now at shallower depths of ~10 km. This activity drastically intensified in the following days and earthquakes migrated W, NW and N, gradually approaching the surface. Ground deformation was detected by 5 IGN GNSS stations, 1 inclinometer and InSAR measurements of the island's SW, suggesting dike intrusions[2,23,24]. On September 19, 2021, seismicity reached very shallow depth and a maximum $M_L$ of 3.1 was widely felt. The eruption started at 14:10 UTC along a NW-SE fissure and remained active for 85 days and 8 hours, to date the longest known eruption at La Palma. The eruption showed variable explosive behaviour[25], with a mean eruptive column height of ~3500 m and emissions of large lava flows (12.19 km²), forming 2 lava deltas on the Western coast[1].

The co-eruptive seismicity[26] (19.09.2021–13.12.2021) included 7232 events with hypocentres distributed at two depth intervals of ~10–14 km and ~33–39 km beneath the central area of Cumbre Vieja; 13 earthquakes reached maximum intensity IV-V (EMS98) and a few were felt even at El Hierro, La Gomera and Tenerife islands. Seismicity reached a peak $M_L$ 4.1 on November 19, 2021. The eruption ended abruptly on December 13 at 22:21 UTC, after two days of intensified volcanic activity, reaching its maximum eruptive column (8500 m height).

Here we use seismological methods to analyse the whole volcanic reactivation providing novel key results. We improved the preliminary IGN catalogue by performing a relative location based on waveform cross correlation. We also provide an unprecedented Moment Tensor (MT) catalogue for La Palma and the whole Canary Islands. Based on

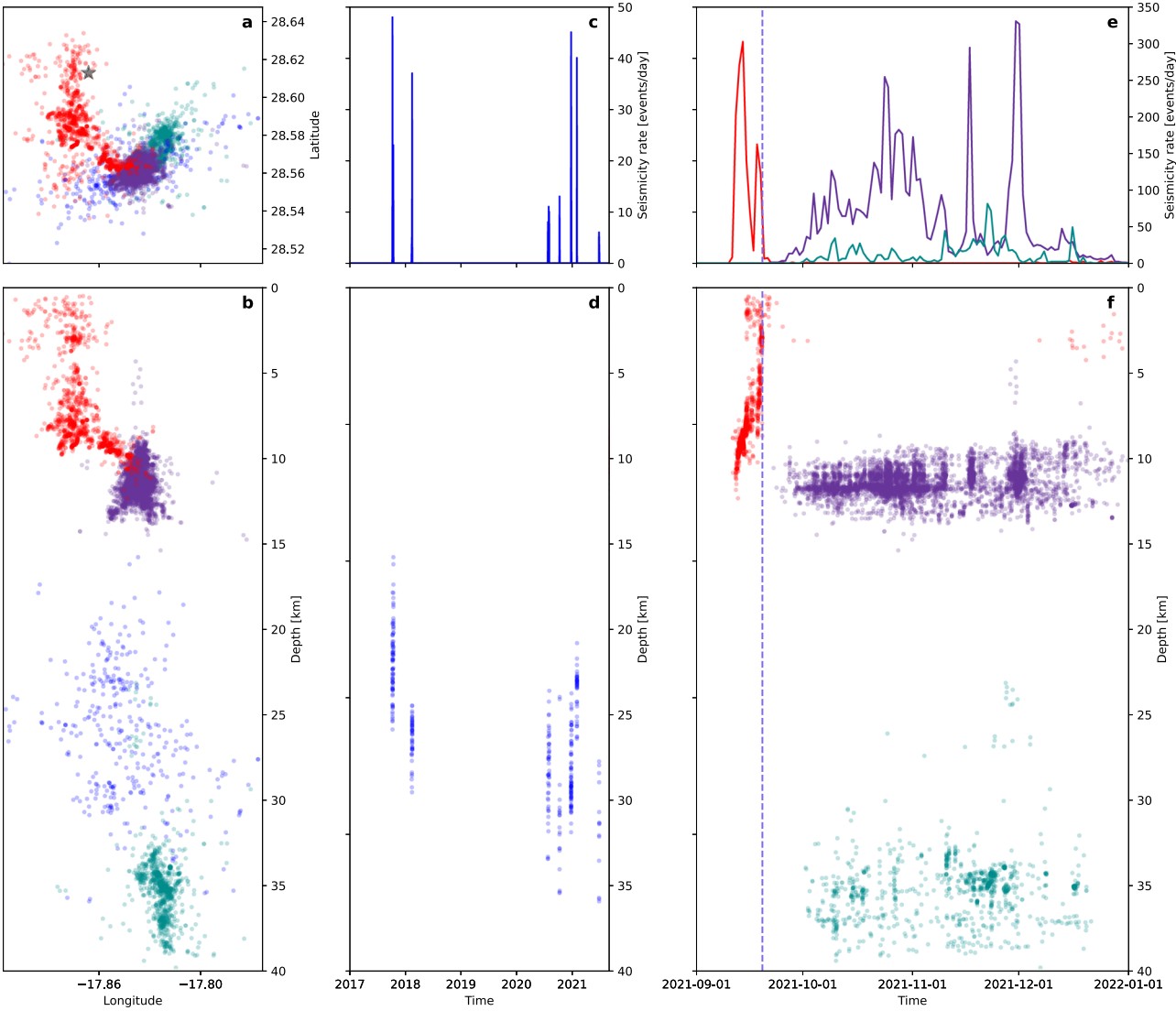

**Fig. 2 | Spatiotemporal evolution of seismicity at La Palma. a, b** Relocated seismicity and location of the first vent (black star) in map view (**a**) and vertically by longitude, where the aspect ratio is conserved (**b**). **c** Seismicity rate during the 2017-2021 swarm period. **d** Depth of 2017-2021 swarm events over time. **e** Seismicity rate during September-December 2021, including dike intrusion, eruptive (a dashed line marks the eruption onset) and post-eruptive periods, and **f** Evolution of the depth of seismicity during the same time period. Red, dark violet, blue and green colour correspond to shallow dike-related seismicity, shallow cluster, 2017–2021 swarms and deep cluster, respectively.

our seismological results and considering geodetic and petrological data, we finally develop a conceptual model of the magma feeding and reservoir system beneath the island.

## Results

### Improvement of the seismic catalogue

The IGN seismic catalogue[26] counts 9989 earthquakes at La Palma from 1.1.2017 to 31.12.2021. The relative relocation of 350 earthquakes constrained the swarms in the pre-eruptive period since 2017 at depths of 15.7-35.9 km (Fig. 2). We used the relocations of 8488 earthquakes of the 2021 volcano-seismic unrest (Supplementary Note 3, Supplementary Dataset 1) to resolve seismicity details in time (Figs. 2 and 3) and space (Figs. 4, 5, Supplementary Fig. 3). The last swarm before eruption started on September 11, 2021. Early earthquakes at ~11 km depth were followed by upward and lateral migration to the eruptive vent (Figs. 2 and 3), triggered along a curved crustal dike path before the eruption (1292 earthquakes). We will use the word cluster to refer to groups of earthquakes, sharing similar properties, such as hypocentral locations, moment tensors or waveforms.

Two large spatial seismicity clusters appeared after the eruption onset. The shallow cluster (6145 earthquakes) has a roughly circular shape, depths of 9.9–12.7 km (90% interval) and a diameter of ~3 km (Figs. 2a, b; 4a). It was the most active one, where the seismicity started on September 27 (Fig. 3b), after a tremor quiescence of ~10 hours (Fig. 3e), with seismicity rates showing intensification in mid-late November. It shows two major seismogenic volumes (Fig. 4a, b) and three spatial subclusters (Supplementary Note 5, Supplementary Fig. 9a, b). The activity in the deep cluster (1054 earthquakes), from 32.8–38.1 km depth, extends ~10 km towards NE. Here we observe a more heterogeneous volume, with a number of small-size seismogenic regions separated by less active ones (Figs. 2b, f; 5a, b). Three main subclusters were found (Supplementary Note 5, Supplementary Fig. 10a, b). The deep cluster activity did not start until October 5, with highest rates in October–November 2021, when it reached the largest earthquakes of the series (Mw > 4.0) (Fig. 3a; Supplementary Fig. 8) and a peak magnitude Mw 4.1 (03.11.2021). Within the deep cluster we identify a number of earthquake doublets (Fig. 3c), here defined as pairs of relatively large earthquakes ($M_L \geq 3.5$) occurring within a short

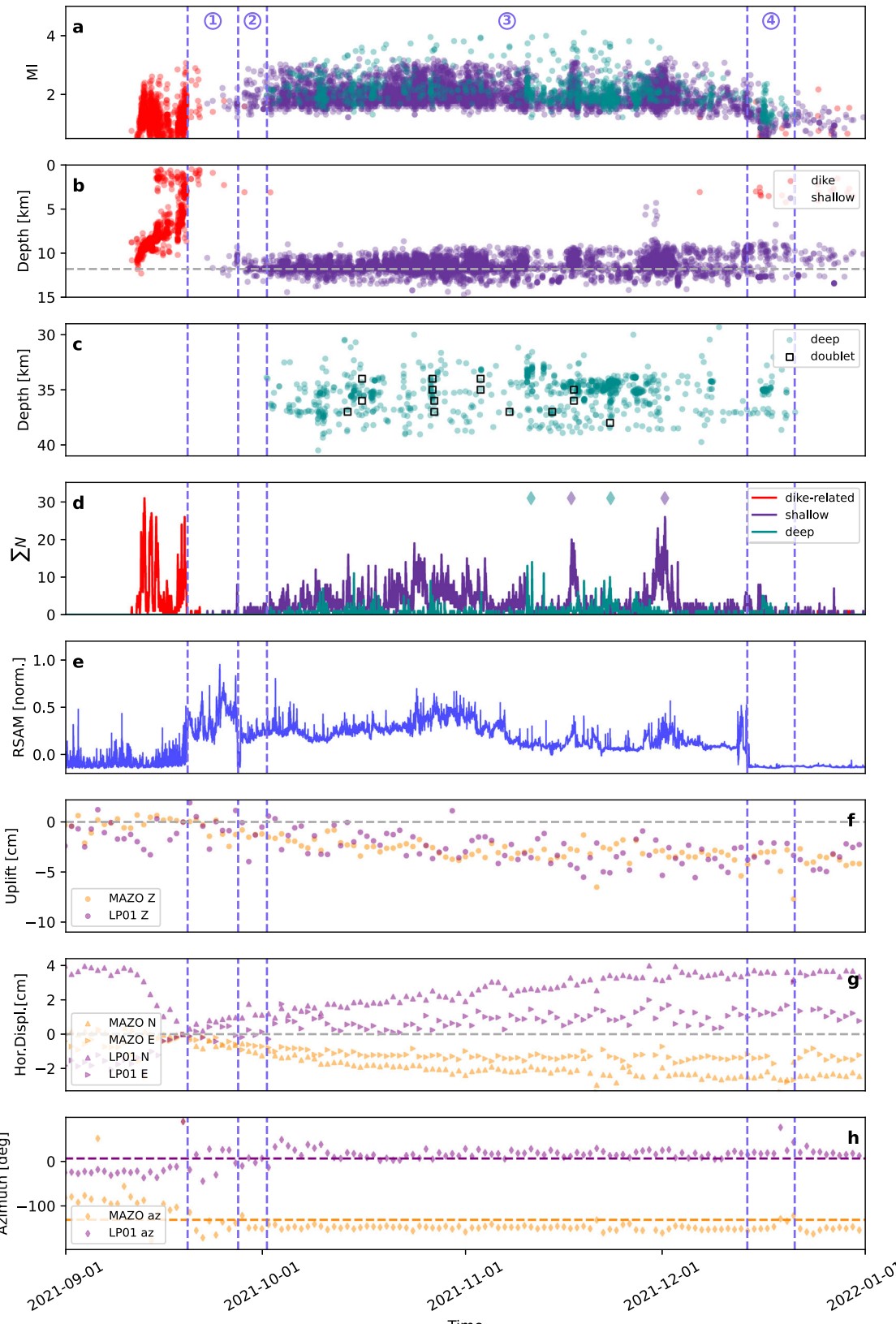

**Fig. 3 | Eruption chronology.** Plots show the temporal evolution of: **a** earthquake magnitudes (red, dark violet and green circles correspond to dike-related, shallow cluster and deep cluster seismicity, respectively, after relocation). **b, c** Hypocentral depths for the different types of seismicity: dike-related shallow events, earthquakes at the shallow and deep cluster, and deep doublets. **d** Daily seismicity rates (dark violet and green diamonds mark temporary increases in the rate of shallow and deep events, respectively). **e** Normalised Real-time Seismic Amplitude Measurement (RSAM) as an indicator for the amplitude of eruption tremor at station CENR. **f**–**h** Vertical and horizontal displacements, and the azimuth of the horizontal displacement as recorded at different GNSS stations. Displacements are shown relative to the eruption time, horizontal dashed lines in panel h indicate the azimuth of the average location of the shallow cluster from the two GNSS stations. The onset and the end of the main stages are marked by blue vertical dashed lines and the blue numbers mark the four main eruption stages.

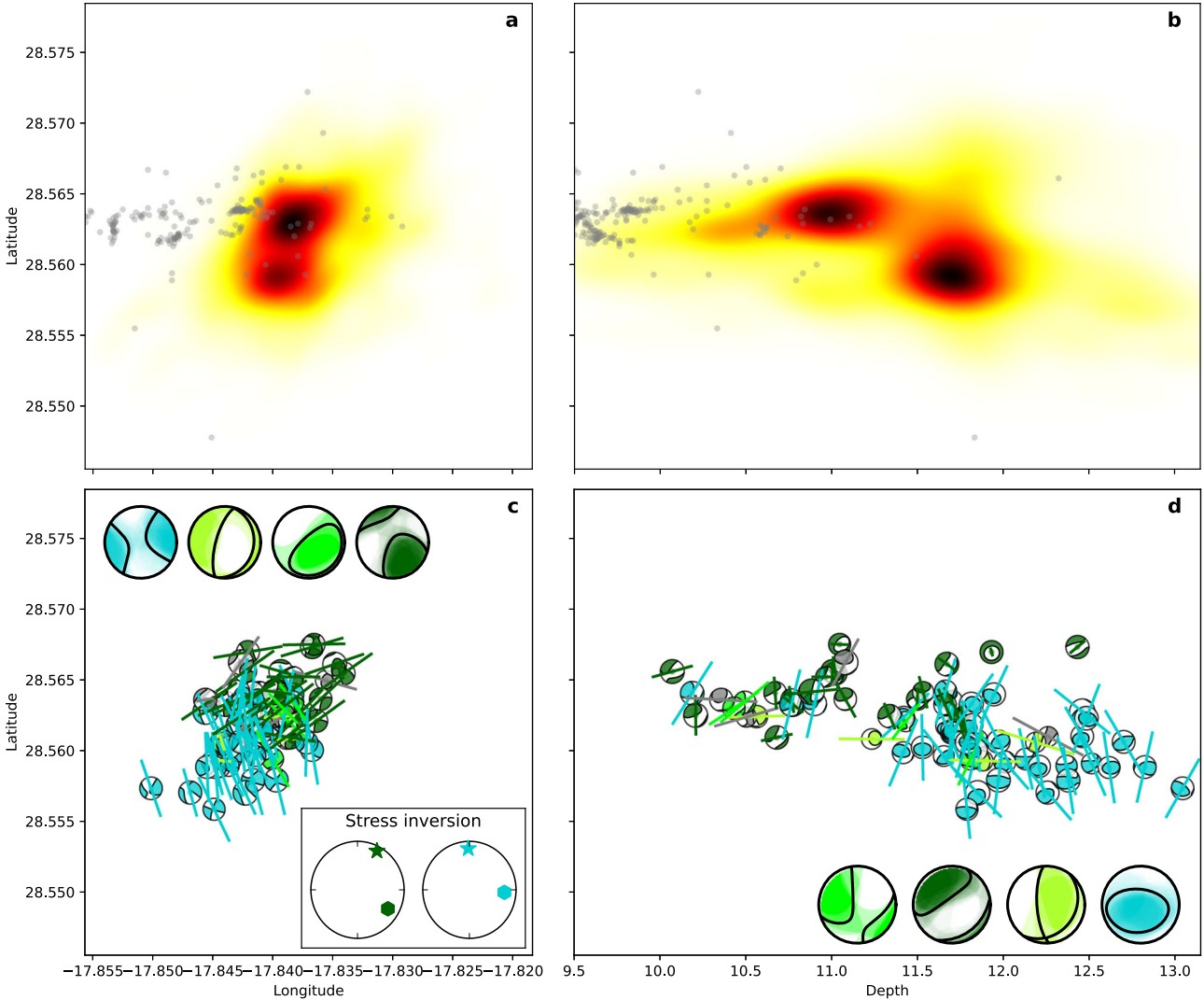

**Fig. 4 | Seismicity distribution at the shallow cluster. a** Normalised density of relocated hypocentres (darker regions corresponding to higher density) in map view and **b** vertically by latitude; hypocentral location of earthquakes associated to the dike intrusion are shown in grey. **c** Moment tensor solutions (small focal spheres) in map view. Focal spheres are coloured according to the moment tensor clustering, with four clusters identified, plus a few unclustered moment tensors (grey). Thick lines show the projection of the corresponding pressure axes. Large focal spheres show the overlay of moment tensors pertaining to each family, with black lines denoting the mean mechanism. Inset shows the stress inversion results ($\sigma_1$ star, $\sigma_3$ hexagon) based on available moment tensors for the two main active regions (panel a). See Supplementary Fig. 12 for $R$-values of the stress inversion. **d** Moment tensor solutions and mean mechanisms for each family are plotted in a cross-section (same as in **b**).

time (from a few seconds to one minute). The shallow and deep co-eruptive cluster were previously identified[27]. However, we located them at larger depths (9–13 km and 33–38 km respectively, compared to[27] 7–11 km and 20–25 km), possibly because of a different velocity model (Supplementary Note 2, Supplementary Fig. 2).

The spatial clustering identifies a few broad regions of seismic activity both at the shallow and deep clusters (Supplementary Note 5; Supplementary Figs. 9a, b and 10a, b). Each of these regions can be further classified based on the waveform similarity. This analysis confirmed the presence of at least 20 families both within the shallow (Supplementary Fig. 9c, d) and the deep seismic clusters (Supplementary Fig. 10c, d). Each of these smaller families display a high waveform similarity (Supplementary Fig. 11), which implies similar locations and similar focal mechanisms[28].

### Moment tensor solutions and stress partitioning
Bodywave MT inversion was obtained for 156 events (Supplementary Note 4; examples and fitting in Supplementary Figs. 4c, d and 5c, d)

sampling both the shallow (73 earthquakes, Supplementary Dataset 2) and deep (83 earthquakes, Supplementary Dataset 3) clusters. Shallow cluster MTs can be classified into four families (Supplementary Fig. 6), the two largest matching well with the two major seismogenic volumes (Fig. 4) and results found by spatial clustering (Supplementary Fig. 9a, b). Focal mechanisms are characterised as oblique to strike-slip, plus a positive compensated linear vector dipole (CLVD) and overall, negative isotropic components. The most populated family (light blue in Fig. 4c, d) is located in the SW and at depths of 11.9 ± 0.6 km; focal planes strike ESE-WNW and NNE-SSW. The second more populated family (dark green in Fig. 4c, d) is shallower (depths 11.2 ± 0.6 km), to the NE, and presents similar focal mechanisms, but with opposite polarities. Hypocentres are not aligned, excluding the activation of a single fault. The two remaining families are smaller and show thrust and normal faulting components (Fig. 4c, d).

At the deep cluster we find similar results (Supplementary Fig. 7) where MTs are also classified into four families, with location and depth shifts and, in some cases, a focal mechanism flip (reversed

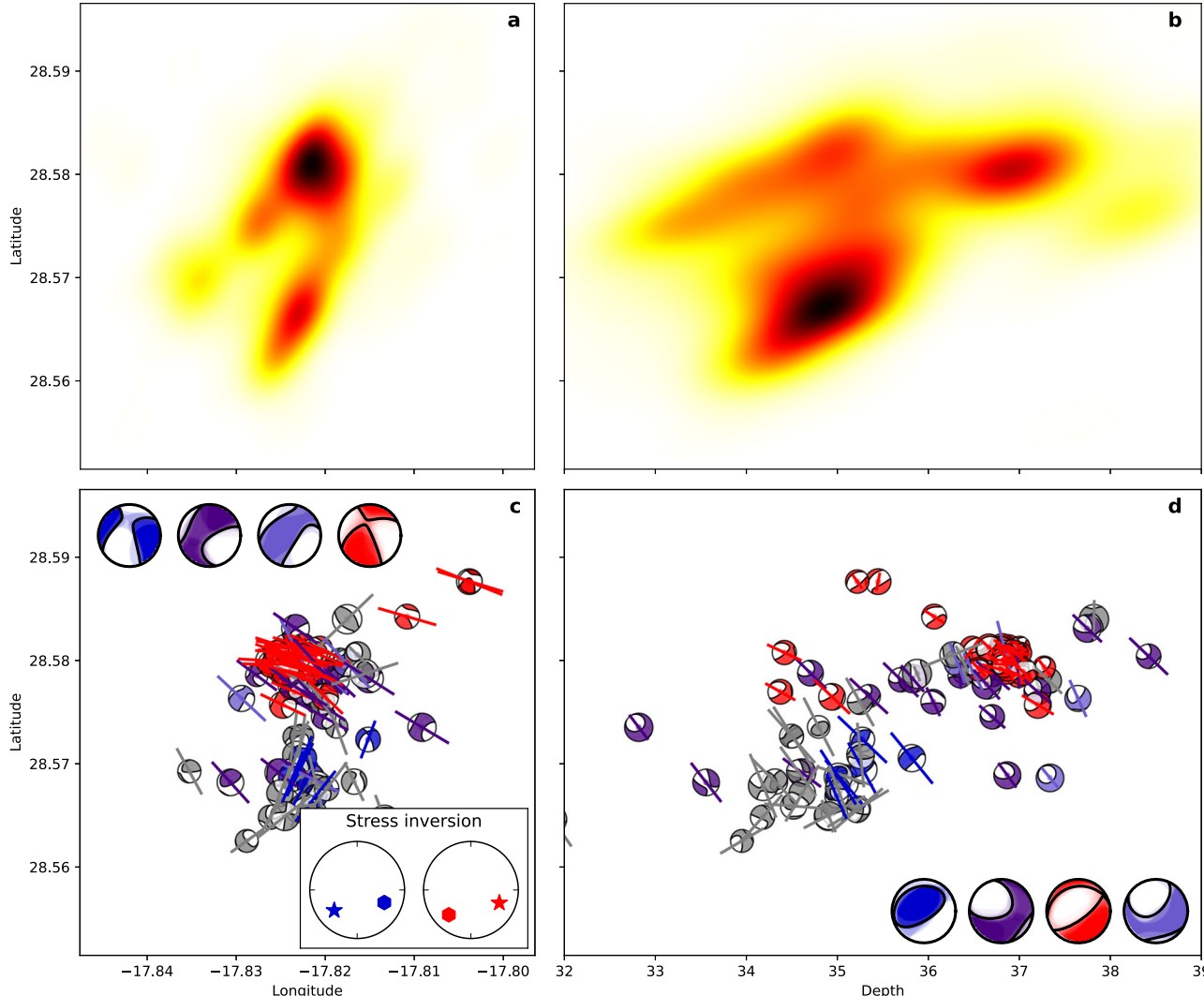

**Fig. 5 | Seismicity distribution at the deep cluster. a** Normalised density of relocated hypocentres (darker regions corresponding to higher density) in map view and **b** vertically by latitude. **c** Moment tensor solutions (small focal spheres) for the moment tensor catalogue as in Fig. 4; the larger families are identified in blue, indigo, violet and red with the rest of the earthquakes in grey. Inset shows the stress inversion results ($\sigma_1$ star, $\sigma_3$ hexagon) based on available moment tensors. **d** Moment tensor solutions and mean mechanisms for each family plotted in cross-section (as in panel **b**).

polarity). The majority of focal mechanisms in the deep cluster are strike-slip to oblique, but others show large thrust or normal components. The family at the top ($35.2 \pm 0.3$ km depth) displays a strike-slip mechanism with tension and pressure axes oriented ESE-WSW and SSW-NNE, respectively (Fig. 5c, d; dark blue), where polarity is reversed compared to the strike-slip mechanism of the largest and deeper ($36.5 \pm 0.8$ km) family (Fig. 5c, d; red). The two remaining families have ESE-WSW pressure axes but include thrust to oblique components. Scattered CLVD and minor isotropic components provide no evidence for any robust non-double couple component (Supplementary Fig. 7).

The 73 and 83 moment tensor solutions of the shallow and deep clusters, which separate families in different depth ranges, were used to estimate the shape and direction of stresses in the upper and deeper volume of the clusters (Supplementary Note 6, Supplementary Fig. 12). Results only roughly confirm the expected stress orientation. At the shallow cluster, the roughly NS oriented compressive stress agrees with the stress orientation inferred from dike paths, while at the deep cluster the NW-SE orientation is slightly rotated compared to the ~N25°W orientation of regional compressive stresses[29]. Results indicate stress rotations in both shallow and deep clusters as a function of depth and time. While the volume of shallow activity is characterised by a horizontal stress system, for the deeper activity cluster there are indications of a transition between strike-slip and normal/thrust faulting.

## Discussion

The evolution of seismicity over ~5 years (Fig. 2) shows a substantial change from the pre- to the co-eruptive phase. We developed a conceptual model that accounts for our observations (Fig. 6). Seven earthquake swarms lasting 2–10 days occurred since 2017, and more frequently from 2020 until August 2021. Most swarm earthquakes were located in the uppermost mantle at 18–32 km depth, coinciding with the depth of pre-eruptive magma storage as inferred from petrological data[7,9]. Changes in gas emissions reported before and after the swarms[19] suggest that seismicity could be caused by transient pressurisations of the magma storage system and intrusions to shallower levels[14,19]. Pressurisation may reflect magma recharge and concomitant exsolution of a $CO_2$-dominated fluid[30,31]. We propose that repeated intrusions since 2017 progressively formed a mushy reservoir in a storage zone that was already hot due to previous volcanic activity[7,9] (Fig. 6a). By September 2021, the reservoir was sufficiently mature and pressurised to cause further magma ascent into the crust

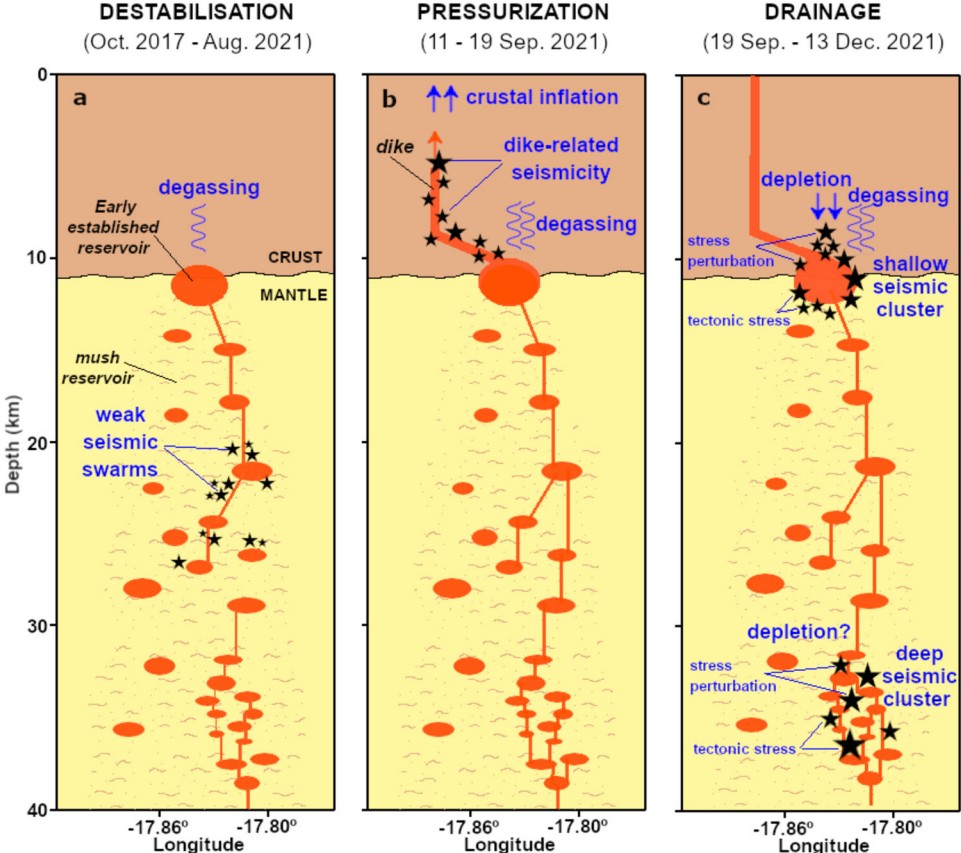

**Fig. 6 | Interpretation of the reactivation process.** Evolution of seismicity and magmatic plumbing at different stages: **a** from October 2021 to August 2021; **b** from 11 to 19 September 2021; **c** during the 2021 eruption. Drainage of the magmatic system was controlled from top to bottom in the early phase of the eruption, and from bottom to top from early November on. Vertical projection by longitude as in Fig. 1b.

and destabilisation of the upper plumbing system. Immediate pre-eruptive seismicity began on 11 September 2021. Progressive migration of seismic foci (Figs. 3 and 4) indicates that dikes propagated from ~12 km depth and reached the surface at *Cabeza de Vaca* (municipality of El Paso) on September 19, at 14:15 UTC[1,23] (Fig. 6b). The curved path of the dikes (Fig. 2a) suggests they propagate through a strong stress heterogeneity. The final N–S trend of the path (Fig. 2a), away from the stress perturbation introduced by the deeper plumbing system, indicates that the background crustal stress at La Palma has the maximum compressive stress $\sigma_1$ oriented ~NS and the minimum principal stress $\sigma_3$ oriented ~EW.

Most of the co-eruptive seismicity occurs within the shallow (9–13 km) and deep (33–38 km) clusters, located above and below the depths of the pre-eruptive swarms (Fig. 2). The lack of earthquakes at ~18–32 km during the co-eruptive period indicates little wall-rock fracturing, probably reflecting that an open magma pathway was established and maintained at high temperature, allowing aseismic magma transfer (Fig. 6c). We denote these three distinct magma storage levels as shallow (9–13 km), intermediate mushy (18–32 km), and deep (33–38 km) reservoir systems. A shear wave velocity anomaly at ~13 km depth was recently resolved by a tomographic study[27], supporting the interpretation of a reservoir at this depth. Our results stress the relevance of the intermediate mushy reservoir for magma storage and the build-up of an eruption, even though it is not mirrored in the co-eruptive seismicity.

The chronology of the co-eruptive seismicity (Fig. 3a–e) has four stages. Stage 1, starting at eruption onset and lasting ~1 week, is characterised by a sharp decrease in seismicity (Fig. 3d) and high amplitude tremor (Fig. 3e). In stage 2, starting September 27, 7:15 UTC when the

tremor amplitude decreased drastically, and remained at minimum level with no magma emission for ~10 h. The shallow cluster seismicity appeared at ~11.8 km depth (Fig. 3b). Simultaneously, subsidence and lateral deformation transients towards the shallow cluster are detected at GNSS stations up to ~15 km (Fig. 3f–h), suggesting that the shallow reservoir started to drain. On October 1 (stage 3), the deep cluster appeared (Figs. 3 and 5). Shallow and deep clusters delimit zones of magma accumulation, providing constraints on their size and shape. At the shallow cluster, seismicity follows a roof-shaped distribution, at or just above the region reached by the magma during the pre-eruptive period. We interpret this as evidence for a volumetric reservoir at ~12 km depth. Our results of pre-eruptive seismicity location suggest this shallow reservoir was established early, although it was silent and undetected before the eruption. Its location, hypothesised by petrological[6,7] and deformation[14] studies, corresponds to the starting point of the dikes that reached the surface (Fig. 2a, b). Early erupted magma, characterised by amphibole in addition to clinopyroxene and olivine macrocrysts, was likely originated at that depth[32,33]. Subsidence (Fig. 3, Supplementary Note 8, Supplementary Fig. 14) and negative isotropic MT components (Supplementary Fig. 6) are consistent with magma drainage from the shallow reservoir and incomplete refill from the mantle reservoirs during stage 3. The deep and more spatially extended cluster is interpreted to reflect a deep, more distributed reservoir system, consisting of multiple magma pockets[7,9]. The ~4-day delayed onset of deep seismicity suggests this was triggered by the partial drainage of the shallow and intermediate reservoirs, resulting in recharge from below. A lower deformation rate by early November may indicate the shallow reservoir was approaching an unstable equilibrium, balancing magma in- and outflow. At the same time, short-

term seismicity rate anomalies alternated at the shallow and deep cluster (Fig. 3d). Seismicity bursts (Fig. 3a–d) were shortly followed by increased lava emission and/or opening of new vents[25,34], which may reflect structural instabilities resulting from increased magma withdrawal from the deep reservoir. An increase of the shallow seismicity rate by late November 2021 (Fig. 3b), reaching 4–8 km depth, may indicate fracturing processes at the roof of the depleting reservoir[35–37]. Stage 4 comprises from the eruption end and the drop of tremor (December 13) to the end of the deep seismicity (December 21). After Stage 4, shallow seismicity became progressively more sporadic.

We have resolved the fine structure of the magma system beneath Cumbre Vieja and discuss its temporal evolution. The upper bound of the shallow seismic cluster outlining a shallow reservoir became progressively shallower during the eruption while leaving a low activity range in its centre (Fig. 2f). The upward progression of seismicity could indicate increased faulting at the reservoir's roof as a consequence of its drainage, a scenario supported[37] by steep tension axes in this upper volume (Fig. 4c, d, light green focal spheres). The decrease of the seismicity rate within the centre of the shallow cluster may be related to changes in the rheological conditions, due to sustained magma flow. Negative isotropic moment tensor components suggest the depletion of the shallow reservoir, confirming the geodetic-based inference of its drainage (Supplementary Fig. 14), and can be related to the high volatile component of the magma in Cumbre Vieja[38]. Indeed, volatiles can explain negative co-seismic isotropic components if adjacent earthquake ruptures weaken the mechanical stability of the reservoir building, and, as a result, reservoir pressure temporarily increases so that the pore volume of the volatiles is co-seismically reduced.

The seismicity chronology (Fig. 3) resolves the dynamic activation of the magmatic system, controlled from top to bottom in the early phase of the eruption. Drainage, strain and pressure loss at the shallow reservoir triggered, with some delay, the drainage of the deeper reservoir (stage 3) and consequently deeper seismicity. Conversely, a bottom to top control was observed from early November when seismicity bursts were followed by increased lava emission. Likewise, after the eruption ended, drainage and seismicity stopped first at depth, and only later at the shallow cluster. The discontinuous evolution of seismicity rates at both clusters suggests discontinuous magma flow, which is also confirmed by variable effusion rates[25,34]. The occurrence of doublets (Fig. 3c), observed for the first time at La Palma, can be interpreted in the frame of a pulse-like magma flow, as doublets may be activated over short time intervals by ascending magma batches.

Shallow cluster seismicity mostly occurs in two regions (Fig. 4, Supplementary Fig. 9) with reversed focal mechanism polarity. Earthquakes located in the bottom part of the cluster have NNW-SSE pressure axes (light blue focal spheres in Fig. 4c, d; Supplementary Fig. 12a), consistent with regional tectonic stresses in the upper crust, which we infer from the N–S orientation of the dike paths away from the reservoir. Conversely, pressure axes of earthquakes at the top of the shallow cluster are rotated by ~90° (dark green focal spheres in Fig. 4c, d; Supplementary Fig. 12b). This striking observation requires a strong stress perturbation, which we attribute to the reservoirs and their drainage. Similar results are found at the deep cluster: again, focal mechanisms of earthquakes located at the bottom of the cluster (red focal spheres in Fig. 5c, d; Supplementary Fig. 12d) have NW-SE pressure axes, roughly consistent with the background stress[29], and those at the top are rotated by ~90° (blue focal spheres in Fig. 5c, d; Supplementary Fig. 12c). This similarity suggests drainage also at the deep reservoir. The observation of ~90° rotated focal mechanisms in volcanic environments[39–42] has been explained by faulting ahead of magma conduit narrowing, failure of solidified basalt plugs[42], intrusion along sub-parallel fractures and movement on small jogs between adjacent dykes[42], but has also been attributed to stress field rotation driven by high-viscous magma flow[40,41], which is not the case of La

Palma[25]. We argue that the focal mechanism reversal is a consequence of local stress heterogeneity introduced by a depleting reservoir (Fig. 6). Our model of stress heterogeneities (Supplementary Note 7) can first explain the curved dike paths when the shallow reservoir is pressurised (Supplementary Fig. 13a). During its depletion we can reproduce higher seismicity towards Northeast and Southwest and reversed focal mechanisms (Supplementary Fig. 13b).

Seismicity, moment tensors and geodetic data provide the key to understanding the behaviour of the complex magma storage system, with a shallow, and intermediate mushy and a more distributed deep reservoir. Our model involves three reservoirs whereas previous interpretations[27] suggested only one large magma reservoir. The 2021 La Palma eruption provides an example of interplay between magma accumulation and withdrawal, pressure increase and release, intermittent magma storage, deformation, seismicity, eruption tremor and effusive lava outflow. Conceptual models obtained from a multi-disciplinary approach, such as the one proposed in this work, help to understand a volcano's behaviour and to improve the early warning and volcano monitoring in oceanic islands.

## Methods

### Hypocentral relocation

A relative relocation of the IGN seismic catalogue since 2017 was performed using a double-difference method, using the HypoDD algorithm[43]. This algorithm improves the hypocentral location by minimising residual travel-time differences of earthquake pairs at a common station with a weighted least square method. Given the significant temporal evolution of the seismic network and the difference among earthquakes and waveform patterns due to variable locations, depths and focal mechanisms over the pre- and co-eruptive sequence, we used different stations for the relocation. For the earthquakes of the pre-eruptive swarms (October 2017 to June 2021), we used seismic records at all stations installed at La Palma and La Gomera at these times. In the case of the shallower seismicity in the weeks preceding the eruption onset (2021.09.11–2021.09.19), as well as for the co-eruptive seismicity located in the shallower cluster, only the stations located on the island of La Palma were used. Finally, for the co-eruptive deeper seismicity we used all stations at La Palma, one at La Gomera, three at El Hierro and four at Tenerife (Supplementary Note 1, Supplementary Fig. 1). In this case, including stations at larger distances was possible due to the larger magnitudes of these earthquakes.

The waveform correlation on P and S phases was performed using time window lengths of 2 and 3 s, respectively. Different pairs were weighted depending on their waveform correlation[44] and only considering waveforms with a correlation of at least 0.75 and corresponding earthquake pairs separated by a maximum distance of 1 km. For time differences based on phase picks, we only considered strongly linked event pairs, with more than 8 common phases and a maximum distance of 3 km between hypocentres. The velocity model used for the relocation was the same as in the IGN catalogue[45] (Supplementary Note 2, Supplementary Fig. 2). Overall, we used 16 million time differences, including ~15 million based on waveform correlation and ~1 million travel-time based on phase picks. On average, we counted with 20 phase data and 2000 time delays per event.

### Moment tensor inversion and stress partitioning

A full moment tensor inversion was performed using the Grond software[46] for earthquakes above a threshold local magnitude of $M_L$ 2.5. We relied on the seismic network by the National Geographic Institute (IGN) of Spain[20], which was complemented by a temporal installation by the GeoForschungsZentrum (GFZ) Potsdam[47]. We use in total 13 seismic stations on La Palma and 6 stations at neighbouring islands up to a distance of 130 km (Supplementary Note 1, Supplementary Fig. 1). The majority of these stations are equipped with broadband seismometers, and a few with short period sensors. Due to the lack of clear surface

waves, the inversion was performed by simultaneously fitting P and S waveforms in the time domain. Displacement seismograms were computed by deconvolving the instrumental response function. Instrument responses were empirically checked by comparison of high frequency teleseismic P phases. Data have been manually revised to exclude single traces in the presence of gaps, large seismic noise, tilt, saturation or overlap with the waveforms of other earthquakes. P and S phases have been manually picked to align waveforms with synthetics. Synthetic seismograms have been computed for a regional velocity model[45] (Supplementary Note 2, Supplementary Fig. 2) using the QSEIS code[48]. The inversion was performed by fitting 1-s time windows on P phase vertical components and S phase transversal components, after applying a 4th order Butterworth bandpass filter between 1 and 5 Hz. The Grond optimization ran over 100,000 iterations, the first 10,000 iterations searching randomly in full parameter space and the later 90,000 progressively increasing the number of searches towards the best fitting solutions. Robust high-quality solutions were found for 156 earthquakes, 73 in the shallow cluster and 83 in the deep cluster. The moment tensor inversion provides the following parameters for each earthquake: centroid depth, centroid time and the 6 independent moment tensor components. Note that the epicentral coordinates of the centroid are fixed to those of the original catalogue and are not inverted, because the inversion setup, which only uses bodywaves and has a manual alignment of synthetic and observed waveforms, has little power to resolve lateral shifts of the location. The so-called best solutions to which we refer above, were obtained using all available data. This best inversion is accompanied by simultaneous inversions using 100 bootstrap chains, where data are differently weighted to provide an ensemble of 100 additional solutions for each earthquake; these solutions can be used to estimate mean solutions and parameter uncertainties. For this application, mean solutions are not substantially different from best ones.

For the stress inversion we used a least squares optimization method (see also Supplementary Note 6 and Supplementary Fig. 12), assuming that slip on an unfavourably oriented fault would occur in the direction of maximal strain energy release. We divide the 73 and 83 moment tensor solutions of the shallow and deep activity cluster, respectively, into spatially concentrated subvolumes, separated by depth, and assume that stress within each subvolume was homogeneous. The method then resolved the direction and relative magnitude of the principal stresses for each subvolume. However, because the stress close to reservoirs and within the plumbing systems are highly perturbed and heterogeneous, the estimated stress tensors and their rotations are approximate. The true stress field is likely even more heterogeneous. To understand the possible reasons and controls of the stress rotations, we employed a simplified analytical model of stress in a tectonic loaded infinite block with a circular, pressurised reservoir at its centre. We have thus developed stress models that can roughly explain the observed stress rotations, the migration path of the eruptive dikes, and the patterns of seismicity clusters at the reservoirs (Supplementary Note 7, Supplementary Fig. 13).

### Seismicity classification

We used a density-based clustering algorithm[49] to classify the relocated seismicity upon the hypocentral location and the moment tensor catalogue upon the similarity of the focal mechanism[50] (Supplementary Note 5). Given the heterogeneity of the seismicity's spatial distribution, the shallow and deep seismogenic volumes were considered separately. For the spatial clustering (Supplementary Figs. 9a, b and 10a, b) we used the following clustering parameters: Nmin = 150, ε = 0.0002 for the shallow cluster, implying a cluster is formed when one hypocentre is surrounded by at least 150 other earthquake within 200 m distance; and Nmin = 50, ε = 0.0004 for the deep cluster, these different values were chosen according to the smaller size and less uniform distribution of the deep seismicity. For the moment tensor clustering

(Figs. 4c, d and 5c, d), we set the clustering parameters to: Nmin = 2, ε = 0.2 for the shallow cluster, where a cluster is formed when for one focal mechanism there are at least 2 more differing by a Kagan angle[50] $k \leq 24°$ and Nmin = 2, ε = 0.15 for the deep cluster.

We also classified the earthquakes based on their full waveform cross correlation (Supplementary Figs. 9c, d and 10c, d)[51]. A high waveform correlation at multiple stations implies both a similar hypocentral location and a similar focal mechanism[52]. This method has been successfully applied to different types of seismic signals at many volcanoes, including volcano-tectonic[51] and long-period events[53,54]. We consider time window lengths of 8 s to ensure that the whole seismic signal of the earthquake at more distant stations are also included. A band-pass filter between 6 and 16 Hz was applied to remove the most energetic band of the eruption tremor. Finally, the classification was performed considering a different station list for each seismogenic volume. For the shallower cluster we used 10 stations in La Palma and one in La Gomera while 4 stations in La Palma, one in La Gomera and one in Tenerife were used for the deepest cluster (Supplementary Fig. 1). In this procedure[51] an average correlation matrix is constructed as a weighted linear combination of the different matrices obtained for each station. While average correlation values may lower those at a single station, this approach strengthens the results by ensuring a simultaneous high similarity at many stations. After a hierarchic analysis we chose a threshold of 0.7 for the correlation factor to create the families in both clusters.

### Eruption tremor

The eruption tremor was recorded at all seismic stations since September 19, with largest amplitudes closest to the eruptive vent. We evaluated the evolution of eruption tremor amplitude using continuous seismic data of station CENR, which belongs to the IGN permanent seismic network and is located closest to the eruption vents. We tracked the temporal evolution of the eruption tremor using the Real-time Seismic Amplitude Measurement (RSAM)[55], which was computed with a dedicated software (ThomasLecocq/ssxm: RSAM/ RSEM - SSAM/SSEM easy code), using 10-min time windows. Time series were normalised by subtracting the median and dividing by the standard deviation.

### Deformation

In order to get daily GNSS time series, GNSS data were processed with Bernese software v5.2[56] considering a regional GNSS network consisting of more than 30 GNSS stations located in the Canary Islands and surrounding areas (Azores, South of Spain and North of Africa). Coordinates were computed in the ITRF2014 reference frame[57] applying ocean-loading model FES2004[58], the IGS (International GNSS Service) absolute antenna phase centre models and IGS satellite orbits.

## Data availability

The seismic catalogue of IGN is available at https://www.ign.es/web/ ign/portal/vlc-catalogo. Data used in this study are hosted at IGN data center and part of them at GEOFON data center. Some of the data are open access and for the remaining station data can be obtained upon request (volcanologia@mitma.es). The advanced data generated in this study (seismic relocated catalogue and moment tensor catalogue at the shallow and deep cluster) are provided as Supplementary Datasets 1–3.

## Code availability

All seismological software used in this work is open source. The codes used to generate individual results are available through the contact information from the original publications. Fig. 1, Supplementary Figs. 1 and 14 were obtained using Generic Mapping Tools (GMT)[59]. Requests for further materials should be directed to C.d.F. (cdelfresno@mitma.es).

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

## Acknowledgements

We are thankful to: T. Walter, N. Richter, A. Shevchenko, E. Zorn, P. J. González, C. C. Rodríguez, J. M. Tordesillas, L. Lozano and the IGN Volcano Monitoring Team for their support in the installation and maintenance of the seismic network; J. Barco, S. Mikulla, M. Kriegerowski and D. Vollmer for preparing seismic stations; A. Geyer for her clarifications on the regional stress field in La Palma; J. Rueda for the $M_L$ values. Temporal deployment of GFZ stations was supported by the GFZ Task Force Project "La Palma". C.d.F. and I.D.C. received funding by the Ministry of Science and Innovation (Spain) within the framework of the project PID2020-114682RB-C32.

## Author contributions

C.d.F. and S.C. contributed equally to the manuscript. C.d.F. and S.C. coordinated the project, conceived the manuscript and figures and analysed, modelled and interpretated seismological data. S.C. performed the moment tensor inversion. I.D.C. performed the relative relocation. E.A.D.S. contributed to the seismic catalogue analysis and clustering. T.D. performed the stress inversion and modelling. L.G.C. processed and analysed GNSS data. C.d.f., S.C., A.K., T.D. and C.L. contributed to the interpretation of results and discussion section. S.M., C.M., C.V.M., R.L.D. are responsible for the installation of temporary network, network maintenance during the eruption and data acquisition. C.d.F. and S.C. drafted the manuscript. All authors reviewed the manuscript.

## Competing interests

The authors declare no competing interests.
