## [Peer Review File · Nature Communications]

REVIEWER COMMENTS

Reviewer #1 (Remarks to the Author):

I have read and reviewed the manuscript "Magmatic plumbing and dynamic evolution of the 2021 La Palma (Spain) eruption" by Del Fresno et al. This study mostly utilizes relocations of seismic events as well as new moment tensor solutions in combination with other seismic and deformational observations to characterize the 2021 La Palma eruption and establish a new conceptual model of melt storage and transport underneath the island. This manuscript tackles a topic which a wide audience will find very interesting and provides the first geophysical evidence for the presence of several melt bodies and their interactions. The seismological methods used are all sound and seem to be applied flawlessly with some minor questions here and there. The manuscript can use some more clarity and streamlining, especially in the results and discussion section, as well as clearer reference to the figures or even reduction of subfigures. Overall, I think this manuscript will add considerably to our knowledge of the La Palma plumbing system and its dynamics.

My 'main' issues are that the reader is left in the dark as to why there the observed earthquakes are clustered based on three different criteria, how they relate to each other and what we can learn from that. Figures 4 and 5 need some work in this context as detailed below and also referenced correctly and more abundantly. As the methods will not appear in the main text, authors needs to make sure they express themselves very clearly as to which clustering is which. Overall, I can only find the MT clustering to be useful at present while the other two are possibly interesting, but no major conclusions are drawn from them.

I would further appreciate it if the authors could make it clear throughout the manuscript what they regard as 'swarm' and what as 'cluster' and if we can learn anything from the difference in behaviour. Also, figures are often cited as a whole and not as a subpanel. While I realize that some of the information is present in many subfigures and that this format is particularly short, I would ask the authors to really think about citing only the relevant subpanels and making it very clear to which one they refer. Overall, I would ask the authors to think about decreasing the information on the individual figure; some panels could be cut or put in the supplements. This would additionally help to streamline the main text.

Line 28: There is a typo in the volume estimate the source you cite say 200 million m³.

Line 93: What do you mean by 'swarms' in this context?

Line 96: 'Activity' is not clear in the context.

Line 97: I don't think the cited figures actually show the migration to the vent, because the vent location is not on the maps. Please add it to the mentioned figures or refer to Fig 1, which shows the vent location + seismicity, not in time but at least in map view.

Line 98: You mention in the introduction that ground deformation starting September 11th suggested "dike intrusions". Here, you now state that seismicity was triggered "long the curved crustal dike path". While I don't want to dispute the fact that seismicity was likely caused by the dike intrusion, your are still in your results section and so far, it has not been clarified whether there was one or several dikes. Please rephrase.

Line 99: I assume you mean "during the eruption" or "after the eruption onset"? Please rephrase.

Line 101: You refer to a pause in seismicity or actually tremor? Or both?

Line 103: Please add "from density cluster analysis" to make it clear to the reader. On that note, I have trouble finding the major two seismogenic regions for the shallow cluster. What I can from Fig 2f is that within the shallow layer, you may have a cluster along both upper and lower depth limits of the volume, as well as what looks like swarm like behaviour between upper and lower depth limits. In Fig 4a, I see three clusters so it's unclear you refer to only two seismogenic volumes. Also the figure caption of 4a doesn't say how these clusters where obtained. If you want to discuss the temporal evolution, refer to Fig 3.

Line 104-106: Here you at least describe all clusters that are present in 5a/b, but I find that at least two of the clusters are overlapping in depth.

Line 112: Please define MT once.

Line 115-122: As said above, it is difficult to see the separate seismogenic volumes in the shallow layer in this figure. Of course I can see the colored clusters, but I am not sure how useful these actually are because you don't seem to refer to them further in the discussion. Also, I was a little lost in this section, because you use the same colors for a/b and c/d even though the basis for clustering is

very different and the results are similar but not the same. I can see on map view a and c are similar, but b and d differ substantially. Also, you have four coloured clusters in c/d and not three as described in the figure caption (in the main text it is correct). I would suggest using different colours between ab and cd.

Line 138-145: I am surprised you give this clustering special mention compared to the density clustering which you only mention in the methods. Also I don't really see how you utilize all these different clustering techniques. It is entirely unclear to me how the sentence "Consequently, we can assume that smaller events within a family also share a similar focal mechanism and thus plot their spatial distribution" actually relates to the figures. To me, this would mean that you check if you have one or more moment tensors available for each of the 20 waveform clusters and if so, assume that those events within the cluster have the same mechanism. But what does that have to do with what you plot? Looks to me as if you simply plot all 20 clusters? Please make this more clear. For the reader, it would possibly make more sense if you restructured your results. First clustering by density, then by waveform and then MTs.

Line 148: How do they differ from clusters? Did you relocate them as well? How do they differ from the published locations?

Line 157: Here you could already refer to your final model.

Line 161: Confusing, please rephrase!

Line 209: 'established early'

Line 229: It would be very useful to the reader to have the effusion rates within Fig 3.

Line 230: What are 'doublets' in the context?

Line 246: The term 'conceptual model' is misleading here as you are referring to an actual stress model, which is not mentioned at all in the short text. Please rework and be precise.

Line 204-249: You start this paragraph to describe your conceptual model, but you consistently drop new information and analyses into the mix while seldom referring to your final sketch. I would either suggest to first, state all what you haven't mentioned yet and then carefully explain your sketch or to sublabel your sketch and devote one paragraph to each of the three panels.

Line 255: You have not mentioned monogenetic volcanism at all before. What is so particular about your observations which is only true for monogenetic cones as compared to i.e. strato-volcanoes or flank eruptions?

Line 328: not sure that 'reservoir stressing models' is the correct term here.

Line 336: Have do these parameters affect the outcome of the cluster analysis? I find the look of i.e. your shallow clusters interesting, as it seems from your plots 4a,b that for the cyan cluster, some events are included even though they are clearly separated by a gap while others which are right next to an event which belongs to the cluster are excluded.

Line 421: 'pressure axes'

Fig 2. Please mark start and end of eruption in thowaw subplots that have a time dependency.

Fig 3. RSAM of which station? Simply put the name in the figure or caption.

Fig 4. In you caption, you mention three clusters which are coloured in turquoise, dark green and light green for subplot a and b. For c and d you colour four moment tensor clusters, with two shades of light green but say they are as described in a. So there is a fourth cluster missing in a and b. e/f I am not sure a plot with 20 clusters is actually useful to look at.

Fig 5. Same remark as Fig 4

Supplementary Material:

Fig 3 -10 The resolution needs to improve.

Fig 3: The scale or legend for the size of the circles/magnitudes is missing. As you are using different km scales for every subplot, you might have to add a legend to every subfigure as the size of the symbols may be different.

Reviewer #2 (Remarks to the Author):

This article describes the seismicity during the pre- and co-eruptive period at Cumbre Vieja volcano in Canary Islands. The backbone of the study is using seismological methods, e.g. relocation, MT inversions, stress inversion, waveform cross correlation. In addition, the results are contrasted with deformation and petrology, achieving an interesting multi-disciplinary conceptual model.

The study is worth of publication since it deals with a recent eruption (original), no previous seismic source studies in the area, it is useful for volcano monitoring, and the methodology selected is among the best to perform such studies. It is also important the multi-disciplinary nature of the study.

I suggest moderate corrections. The list of comments is long, but more about the presentation of results and clarifications rather than criticise the science of the study.

Abstract: No mention to the data or the provider (I understand Instituto Geofisico Nacional).
line 19: mechanism flip, are those DC or non-DC solutions? I think it is interesting to mention here to catch attention on the source mechanism.

Introduction:

line 27: remove among

line 28-31: Separate sentence in two. Also, there are two articles in lines 51-52 that are referred to swarms, why these do not count? Explicit previous seismology articles and petrological. Gravity was not mentioned in the whole article, I don't see the point mentioning it.

line 31-32: remove ", which" and add "they" before "provide"

line 36: "passive margin" - is it accurate enough in geological terms? What about South Africa?

line 41: "flank collapses" - provide citation

line 58: - remove comma after isotopic ratio

line 56-60: Sentence too long, is it really needed for the article? To my understanding, it does not have implications in the conceptual model or discussion.

line 64: define IGN. First in the abstract and then here again.

line 66: There are reported ML, mbLg, Mw. Do they all match? Use only one for simplicity. ML never reaches 4.0, but the MT inversion methodology mention a minimum magnitude of 4 to invert.

lines 54-81: Both paragraphs are a bit messy, I'd start with network enhancements and define IGN. Then the general chronological evolution of the pre-eruptive in one paragraph and the co-eruptive in another.

Results:

line 96: "Eruptive activity started...". Seismic activity already started by September 2021

line 98: a instead of the

line 99: "after the eruption onset" or "during the eruption"

line 101: Fig 2, right? ... also, "the seismicity started on September..."

line 102-103: I can see 3 clusters in Fig 4a-b

line 101-108: Text jumps from shallow cluster to deep back and forth. Needs more organisation.

line 107-108: You mentioned this largest earthquake on November 19 and ML 4.1, is this the same? Consistency in the magnitudes

line 109: Mw>4, how they relate do ML, in Fig 3a there are hardly events larger than ML 4. Again, Consistency in magnitudes.

Moment tensor solutions and stress partitioning

line 112: Fig 4c and 5c, examples and fitting in the suppl 4-5

line 114: Classification is confusing, I suggest to find the most meaningful and stick only with one, i.e. locations, MT or waveforms. Usually in volcano seismology is used the waveforms, but in this case is not very informative, I'd suggest MT.

line 116: oblique means normal or thrust?

line 117-118: first, second family are not defined in Fig 4, they are color-coded, please just use one classification.

line 119: 11.3 km seems a bit off (deepest point instead of an average). This happened with several clusters and depths, revise them all.

Line 122: When changing from shallow to deep cluster, start a new paragraph.

line 124: There are shallow and deep clusters, and then shallowest/deepest deep cluster. Find another notation, refer to the Figures and colours.

line 128: Scattered CLVD, do you mean in the Hudson plot? If so, it is Suppl Fig 7

line 133: Results show SSW-NNE pressure axis, the regional NNW-SSE. To me they don't match, do they match within errors? If so, quantify.

line 134-136: In Suppl Fig 9, I can see Sigma 1 and 3 subhorizontal for shallow. Deep events, instead

of shallow, have certain migration to the centre of the stereographic net, which I understand is a sign of normal/thrust behaviour.

Family identification and clustering

line 140-142: You need to support your claims with visuals. I can read 3 and 2 families, but in Fig 4e-f and 5e-f I cannot understand which ones are the ones you are referring. I understand you made effort in this, but this section just confuses.

Discussion

Define pre-eruptive and co-eruptive in the introduction. Maybe define different background colours in Fig 3 for pre- and co-eruptive.

line 157: replace summer by a month/day.

line 168: Personal preference: use UTC.

line 174: UTC time

line 177: Results from GNSS maybe using arrows in the map (Fig 1)

line 186: ...was likely originated...

line 187: Suppl Fig 6

line 195: remove alternatively

line 196: seismicity rate anomalies, refer to a figure and indicate it. I can see seismicity rate in Fig 2e and 3d, but there are several peaks. Indicate time between responses.

line 196: You mention several times the lava emissions and seem to have a non-trivial role in the conceptual model. I request to include lava emissions in Fig 3 if you can get the data.

line 199: "by late November"

line 208: "Our results". Your results cannot provide this evidence. Maybe petrological/geochemical, but if you don't have seismicity (silent), you cannot claim that.

line 214: Fig 4d, are you referring to the dark green beachball? mention the color you are referring.

line 226: lava emission

line 229: lava emission

line 232: citation for magma pulses producing doublets

line 234: shallow cluster seismicity in two regions. I saw 3 colours in Fig 4 a-b. Refer to colours if you are grouping them.

line 235: pressure axis

line 237-240: repetition of ideas. Also, mention the shear stress pattern change that enable the curved structure observed. The deep cluster seems to have the rotation in the opposite direction, when compared with the Suppl Fig 10.

line 247: "curved path" and relationship with Fig 10a. The locations follow the enhanced shear stress areas for the case of N-S background stress. Fig 4c and 5c show NNW-SSE orientation of the background stress. How this will affect the results? Is it within the errors?

line 255: why monogenetic only?

Methods

line 279: Velocity structure: How does the velocity structure affect the shallow locations? How the residuals behave?

I can see more dispersion in shallow seismicity (depths ~3km). Can it be due to constant properties in quite a thick layer (0-5 km)?

Shallow <5 km seismicity is not discussed in the article. Shallow seismicity provides critical insight for monitoring purposes when the crisis is imminent.

Moment tensor inversion and stress partitioning

line 286: mbLg was not mentioned in the main body of the article. Also, I couldn't see events with magnitude larger than 4 in Fig 3a. Which deep events were used? Is the plot 3a missing larger events?

line 299: not local, regional.

Seismicity classification

I suggest to use one classification. It is confusing to use 3 different criteria.

Figure 1:

Larger fonts in the inset

Figure 3

Keep the Y-labels aligned.

No $ML > 4$ for MT inversion?

3d - indicate the relationship between deep and shallow bursts with arrows.

Use stages or phases, but no both. I suggest stages to avoid confusion with P and S phases.

Figure 4:

light green beachball in plan view and profile is not consistent.

Shallow cluster was shown in purple-ish in Figs 1,2 and 3. Keep consistent.

Use one clustering criteria, this Figure is extremely confusing.

line 420: "identified as in a". This Figure has 4 beachballs and 3 colours in a. Not consistent.

line 421: pressure axes

Figure 5:

Deep cluster was shown in green-ish in Figs 1,2 and 3. Keep consistent.

Figure 6:

indicate the curved dyke propagation with arrows.

Supplementary material:

Fig 4: Example of inversion of a 3.6 event, in the text says inversions are carried out for events with magnitudes above 3.8 for shallow cluster

Fig 8: Why not show the stacked waveforms?

Fig 9: I can see a more important normal/thrust behaviour in the deep cluster, contrary to what the text says.

line 89: Definition of R equal to 1.

Fig 10: the locations fit with the numerical simulation when the pressure axis is N-S, which is not the real case. There must be a rotation ~ 30 degrees

Rodrigo Contreras-Arratia

RESPONSE TO REVIEWERS

In the following paragraphs, we reply to the reviewer comments. We mark in black the comments by reviewers, and in blue our reply. Major changes to our original submission concern two main questions raised by both reviewers: seismicity clustering and streamlining our results. We first address these two issues, and then provide a point-to-point reply to all comments.

1. Seismicity clustering

Seismic clustering has been used to discuss the geometry of active seismogenic structures and the heterogeneity and rotation of focal mechanisms. We recognize, however, that former Figs. 4 and 5 were too dense and confusing, and that not all results shown there were discussed.

We decided to simplify the two figures, and now only show two rows of panels (in map view and cross section), which are illustrative of the most important results. The upper one is illustrating the density of seismicity and it is useful to appreciate the spatial distribution of seismicity. The lower one is dedicated to focal mechanisms, and it is almost unchanged from the previous version.

Former results of the spatial clustering, and the more accurate classification based on waveform correlation, have been removed from the main text and included as new figures in the supplementary material.

2. Streamlining main text and results

We revised the main text and, specifically, the results and discussion sections, to streamline the main messages of our manuscript. The revised text now provides a clearer presentation of our most important results. Reworking figures 4 and 5 (see main comment #1) was also accompanied by a revision of the corresponding text, which is now more fluent and to the point.

REVIEWER COMMENTS

Reviewer #1 (Remarks to the Author):

I have read and reviewed the manuscript “Magmatic plumbing and dynamic evolution of the 2021 La Palma (Spain) eruption” by Del Fresno et al. This study mostly utilizes relocations of seismic events as well as new moment tensor solutions in combination with other seismic and deformational observations to characterize the 2021 La Palma eruption and establish a new conceptual model of melt storage and transport underneath the island. This manuscript tackles a topic which a wide audience will find very interesting and provides the first geophysical evidence for the presence of several melt bodies and their interactions. The seismological methods used are all sound and seem to be applied flawlessly with some minor questions here and there. The manuscript can use some more clarity and streamlining, especially in the results and discussion section, as well as clearer reference to the figures or even reduction of subfigures. Overall, I think this manuscript will add considerably to our knowledge of the La Palma plumbing system and its dynamics.

We are thankful for the appreciation of our work, the careful review and the constructive comments.

My ‘main’ issues are that the reader is left in the dark as to why there the observed earthquakes are clustered based on three different criteria, how they relate to each other and what we can learn from that. Figures 4 and 5 need some work in this context as detailed below and also referenced correctly and more abundantly. As the methods will not appear in the main text, authors needs to make sure they express themselves very clearly as to which clustering is which. Overall, I can only find the MT clustering to be useful at present while the other two are possibly interesting, but no major conclusions are drawn from them.

We considered and replied to these main issues in the answers to main comments #1 and #2 (see above). We improved the figures 4 and 5 and simplified the discussion on clustering results, shifting some of the clustering results to the Supplementary Material. We also simplified and streamlined the main text.

I would further appreciate it if the authors could make it clear throughout the manuscript what they regard as 'swarm' and what as 'cluster' and if we can learn anything from the difference in behaviour.

We consider as a 'swarm' a series of earthquakes occurring closely in time within a small area and displaying a similar range of magnitudes, with the temporal evolution of magnitudes differing from the typical mainshock-aftershocks pattern. This type of behavior is usually related to very heterogeneous material and non-uniform stress in the seismic source. We modified the manuscript to ensure the term 'swarms' is only used for the seven seismic series that occurred before September 2021 at intermediate depths.

On the other hand, we use the wording 'cluster' for groups of earthquakes which share similar properties (for example, they are closely located in space or they have similar moment tensors).

We cleaned the manuscript, ensuring the wording cluster is only used for the co-eruptive seismicity, where we observe different spatial, waveform and moment tensor clusters.

Also, figures are often cited as a whole and not as a subpanel. While I realize that some of the information is present in many subfigures and that this format is particularly short, I would ask the authors to really think about citing only the relevant subpanels and making it very clear to which one they refer. Overall, I would ask the authors to think about decreasing the information on the individual figure; some panels could be cut or put in the supplements. This would additionally help to streamline the main text.

We improved the manuscript as suggested (see also replies to main comments #1 and #2):

- we checked and improved the figure citations over the whole text.
- we simplified figures 4 and 5.
- we shifted some material to the supplement.
- we cleaned and better focus the text.

Line 28: There is a typo in the volume estimate the source you cite say 200 million m³.
Corrected.

Line 93: What do you mean by 'swarms' in this context?
See answer above.

Line 96: 'Activity' is not clear in the context.
We changed it to "The last swarm before eruption"

Line 97: I don't think the cited figures actually show the migration to the vent, because the vent location is not on the maps. Please add it to the mentioned figures or refer to Fig 1, which shows the vent location + seismicity, not in time but at least in map view.
We added the vent location to Figure 2a.

Line 98: You mention in the introduction that ground deformation starting September 11th suggested "dike intrusions". Here, you now state that seismicity was triggered "along the curved crustal dike path". While I don't want to dispute the fact that seismicity was likely caused by the

dike intrusion, you are still in your results section and so far, it has not been clarified whether there was one or several dikes. Please rephrase.

We homogenized the text to 'dikes' in plural form, based on previous publications.

Line 99: I assume you mean "during the eruption" or "after the eruption onset"? Please rephrase.
Rephrased.

Line 101: You refer to a pause in seismicity or actually tremor? Or both?

There was a clear pause in the tremor signal, as well as a pause in lava and pyroclastic emission. Seismicity in this time was low and weak and not locatable, but not completely absent. In the text we prefer to highlight the clear modulation of the tremor.

Line 103: Please add "from density cluster analysis" to make it clear to the reader. On that note, I have trouble finding the major two seismogenic regions for the shallow cluster.

We modified the text, citing now the new Fig. 4a-b, where the spatial distribution of seismicity and the presence of sub-clusters can be well identified.

What I can from Fig 2f is that within the shallow layer, you may have a cluster along both upper and lower depth limits of the volume, as well as what looks like swarm like behaviour between upper and lower depth limits. In Fig 4a, I see three clusters so it's unclear you refer to only two seismogenic volumes. Also the figure caption of 4a doesn't say how these clusters were obtained. If you want to discuss the temporal evolution, refer to Fig 3.

We simplified the figures and the text. Fig. 2f aims at showing the temporal evolution of the shallow cluster, but not appropriate to see sub-clusters. Conversely, the new Fig. 4a-b shows the major sub-clusters. Further details of the spatial structure are shown in Suppl. Fig. 9.

Line 104-106: Here you at least describe all clusters that are present in 5a/b, but I find that at least two of the clusters are overlapping in depth.

We simplified Fig. 5 in the same way as Fig. 4 and added Suppl. Fig. 10. We improved the text to explain the spatial heterogeneity of seismicity in the deep cluster, providing proper references to the new figures. Indeed, some clusters overlap in depth.

Line 112: Please define MT once.

We included it in the Introduction.

Line 115-122: As said above, it is difficult to see the separate seismogenic volumes in the shallow layer in this figure. Of course I can see the colored clusters, but I am not sure how useful these actually are because you don't seem to refer to them further in the discussion. Also, I was a little lost in this section, because you use the same colors for a/b and c/d even though the basis for clustering is very different and the results are similar but not the same. I can see on map view a and c are similar, but b and d differ substantially. Also, you have four coloured clusters in c/d and not three as described in the figure caption (in the main text it is correct). I would suggest using different colours between ab and cd.

Figures and text on the clustering have been improved (see reply to main comment #1).

Line 138-145: I am surprised you give this clustering special mention compared to the density clustering which you only mention in the methods. Also I don't really see how you utilize all these different clustering techniques. It is entirely unclear to me how the sentence "Consequently, we can assume that smaller events within a family also share a similar focal mechanism and thus plot their spatial distribution" actually relates to the figures. To me, this would mean that you check if you have one or more moment tensors available for each of the 20 waveform clusters and if so, assume that those events within the cluster have the same mechanism. But what does that have to do with

what you plot? Looks to me as if you simply plot all 20 clusters? Please make this more clear.

We agree with the reviewer and improved/simplified the text. Indeed, a high waveform similarity implies a similar focal mechanism within each of the 20 clusters. However, focal mechanisms are only available for some of these clusters, and the aim of the figure is only to show their spatial distribution.

For the reader, it would possibly make more sense if you restructured your results. First clustering by density, then by waveform and then MTs.

We restructured the presentation of the clustering results (see main comment #1).

Line 148: How do they differ from clusters?

We clarified in the text the difference among swarms and clusters.

Did you relocate them as well?

Yes, now we mention it in the results section.

How do they differ from the published locations?

The relocated seismicity better constrains the swarm depths, as mentioned in the text.

Line 157: Here you could already refer to your final model.

Corrected.

Line 161: Confusing, please rephrase!

Rephrased and moved to the paragraph of co-eruptive seismicity.

Line 209: 'established early'.

Corrected and moved to the previous paragraph.

Line 229: It would be very useful to the reader to have the effusion rates within Fig 3.

We do not have the full data to plot the timeline of the effusion rates. Cited references confirm the variability of the rates, but do not provide reliable time series.

Line 230: What are 'doublets' in the context?

We now explicitly declare how we define doublets in the result section (pairs of earthquakes with $M \geq 3.5$ within one minute).

Line 246: The term 'conceptual model' is misleading here as you are referring to an actual stress model, which is not mentioned at all in the short text. Please rework and be precise.

Corrected.

Line 204-249: You start this paragraph to describe your conceptual model, but you consistently drop new information and analyses into the mix while seldom referring to your final sketch. I would either suggest to first, state all what you haven't mentioned yet and then carefully explain your sketch or to sublabel your sketch and devote one paragraph to each of the three panels.

We restructured the whole paragraph, following the reviewer's suggestion.

Line 255: You have not mentioned monogenetic volcanism at all before. What is so particular about your observations which is only true for monogenetic cones as compared to i.e. strato-volcanoes or flank eruptions?

In fact, any multidisciplinary approach can help to understand a volcano's behavior and improve early warning and monitoring. We have removed the word "monogenetic" from the sentence.

Line 328: not sure that 'reservoir stressing models' is the correct term here.

Corrected.

Line 336: Have do these parameters affect the outcome of the cluster analysis? I find the look of i.e. your shallow clusters interesting, as it seems from your plots 4a,b that for the cyan cluster, some events are included even though they are clearly separated by a gap while others which are right next to an event which belongs to the cluster are excluded.

We shifted these results to the supplement (see reply to main comment #1). The choice of the clustering parameters is subjective and can affect the clustering results, in terms of number of clusters, cluster heterogeneity and proportion of unclustered events (see e.g. Cesca 2020). In the spatial clustering application, our choice aimed at the identification of a relatively small number of clusters and leaving a small number of unclustered events.

Line 421: 'pressure axes'.

Corrected.

Fig 2. Please mark start and end of eruption in thow subplots that have a time dependency.

We improved Fig. 2e-f.

Fig 3. RSAM of which station? Simply put the name in the figure or caption.

Done

Fig 4. In you caption, you mention three clusters which are coloured in turquoise, dark green and light green for subplot a and b. For c and d you colour four moment tensor clusters, with two shades of light green but say they are as described in a. So there is a fourth cluster missing in a and b. e/f I am not sure a plot with 20 clusters is actually useful to look at.

Fig 5. Same remark as Fig 4

Figs. 4 and 5 have been restructured and Suppl. Figs. 9 and 10 added. In these elaborations we considered all the reviewer's suggestions.

Supplementary Material:

Fig 3 -10 The resolution needs to improve.

The resolution of topography in panels Fig. 3c,d may not look optimal, but this is because of the small spatial extent of these areas and we do not have access to other DEMs. We still find the figure more informative than without the topography. Fig. 10 seems good.

Fig 3: The scale or legend for the size of the circles/magnitudes is missing. As you are using different km scales for every subplot, you might have to add a legend to every subfigure as the size of the symbols may be different.

Done.

Reviewer #2 (Remarks to the Author):

This article describes the seismicity during the pre- and co-eruptive period at Cumbre Vieja volcano in Canary Islands. The backbone of the study is using seismological methods, e.g. relocation, MT inversions, stress inversion, waveform cross correlation. In addition, the results are contrasted with deformation and petrology, achieving an interesting multi-disciplinary conceptual model.

The study is worth of publication since it deals with a recent eruption (original), no previous seismic source studies in the area, it is useful for volcano monitoring, and the methodology selected is among the best to perform such studies. It is also important the multi-disciplinary nature of the study.

I suggest moderate corrections. The list of comments is long, but more about the presentation of results and clarifications rather than criticise the science of the study.

We are thankful to the reviewer for the appreciation to our work. We carefully considered all reviewer's comments in the revised manuscript and we provide a reply below.

Abstract:

No mention to the data or the provider (I understand Instituto Geofísico Nacional).

line 19: mechanism flip, are those DC or non-DC solutions? I think it is interesting to mention here to catch attention on the source mechanism.

We do not think the data provider should be cited in the abstract, note that there is a strict limitation on the abstract length, as we ensure proper data citation with the manuscript. We followed the suggestion on the moment tensor.

Introduction:

line 27: remove among.

Done.

line 28-31: Separate sentence in two. Also, there are two articles in lines 51-52 that are referred to swarms, why these do not count? Explicit previous seismology articles and petrological. Gravity was not mentioned in the whole article, I don't see the point mentioning it.

We followed the suggestion and splitted the sentence. Here, we aim at providing references to studies on the magmatic feeding system. While gravity is not further discussed in our study, the reference may be of interest for the reader. As for previous seismicity studies, cited later in the text, they were either focusing on a broader region or discussing the seismicity patterns, but not towards the geometry of the plumbing system. We think it would be misleading to cite them here.

line 31-32: remove ", which" and add "they" before "provide".

Done.

line 36: "passive margin" - is it accurate enough in geological terms? What about South Africa

We specified the NW margin.

line 41: "flank collapses" - provide citation

Done

line 58: - remove comma after isotopic ratio.

Done

line 56-60: Sentence too long, is it really needed for the article? To my understanding, it does not have implications in the conceptual model or discussion.

We have decided to leave the sentence in the manuscript as it explains the early geochemical evidence that the 2017 and 2018 swarms were related to a magma intrusion at depth.

line 64: define IGN. First in the abstract and then here again.

In the revised text we define IGN the first time it is mentioned.

line 66: There are reported ML, mbLg, Mw. Do they all match? Use only one for simplicity. ML never reaches 4.0, but the MT inversion methodology mention a minimum magnitude of 4 to invert. We simplified the text, referring to ML, except for the moment tensor analysis which provides Mw estimates. A new Supplementary Figure was added showing the relation between Mw and ML (Suppl. Fig. 8).

lines 54-81: Both paragraphs are a bit messy, I'd start with network enhancements and define IGN. Then the general chronological evolution of the pre-eruptive in one paragraph and the co-eruptive in another.

Corrected.

Results:

line 96: "Eruptive activity started...". Seismic activity already started by September 2021.

Rephrased.

line 98: a instead of the

Done.

line 99: "after the eruption onset" or "during the eruption".

Corrected.

line 101: Fig 2, right? ... also, "the seismicity started on September..."

Yes, Fig 2 and 3a,b. Corrected.

line 102-103: I can see 3 clusters in Fig 4a-b.

Figures 4 and 5 have been improved to clarify (see reply to main comment #1).

line 101-108: Text jumps from shallow cluster to deep back and forth. Needs more organisation.

Reordered.

line 107-108: You mentioned this largest earthquake on November 19 and ML 4.1, is this the same? Consistency in the magnitudes.

It is not the same earthquake. An Mw=4.0 was obtained for the largest local magnitude (19/11/2021 01:08 UTC, ML=4.1), while the maximum value of Mw=4.1 was obtained for the earthquake on 03/11/2021 07:27 UTC, ML=3.9. We modify the text to clarify.

line 109: Mw>4, how they relate do ML, in Fig 3a there are hardly events larger than ML 4. Again, Consistency in magnitudes.

A new figure has been added to Supplementary Material (Suppl. Fig 8) showing the relation between ML and Mw.

Moment tensor solutions and stress partitioning

line 112: Fig 4c and 5c, examples and fitting in the suppl 4-5
Done.

line 114: Classification is confusing, I suggest to find the most meaningful and stick only with one, i.e. locations, MT or waveforms. Usually in volcano seismology is used the waveforms, but in this case is not very informative, I'd suggest MT.

We simplified the clustering results, shifting part of these results to the supplement (see reply to main comment #1).

line 116: oblique means normal or thrust?

We state 'oblique' without further specifications, because both normal and thrust components are present.

line 117-118: first, second family are not defined in Fig 4, they are color-coded, please just use one classification.

We referred to the first and second more populated families, the sentence has been rephrased to clarify.

line 119: 11.3 km seems a bit off (deepest point instead of an average). This happened with several clusters and depths, revise them all.

Checked.

Line 122: When changing from shallow to deep cluster, start a new paragraph.

Corrected.

line 124: There are shallow and deep clusters, and then shallowest/deepest deep cluster. Find another notation, refer to the Figures and colours.

Rephrased and added colors to figure references.

line 128: Scattered CLVD, do you mean in the Hudson plot? If so, it is Suppl Fig 7

Corrected.

line 133: Results show SSW-NNE pressure axis, the regional NNW-SSE. To me they don't match, do they match within errors? If so, quantify.

line 134-136: In Suppl Fig 9, I can see Sigma 1 and 3 subhorizontal for shallow. Deep events, instead of shallow, have certain migration to the centre of the stereographic net, which I understand is a sign of normal/thrust behaviour.

We modified the paragraph to account for both suggestions. In addition, the revised manuscript now better discusses the reliability of reference regional stress orientations beneath La Palma.

Family identification and clustering

line 140-142: You need to support your claims with visuals. I can read 3 and 2 families, but in Fig 4e-f and 5e-f I cannot understand which ones are the ones you are referring. I understand you made effort in this, but this section just confuses.

We have rephrased the section for clarity.

Discussion

Define pre-eruptive and co-eruptive in the introduction. Maybe define different background colours in Fig 3 for pre- and co-eruptive.

Terms “pre-eruptive” and “co-eruptive” are now defined in the introduction.

We tried changing background colors in Fig 3, however this hindered the data of the co-eruptive period, which is a main feature discussed in this figure. Therefore, we chose to leave the original background.

line 157: replace summer by a month/day.

Corrected.

line 168: Personal preference: use UTC.

Corrected.

line 174: UTC time

Corrected.

line 177: Results from GNSS maybe using arrows in the map (Fig 1)...

Fig 1 shows the seismotectonic setting and data used in our work. Results are shown in Figures 2-5 and GNSS results are shown as time series in Fig. 3f,g,h. As it is not possible to add this information to the other main figures in the article, we included a new Supplementary Figure (Suppl. Fig 14).

line 186: ...was likely originated...

Corrected.

line 187: Suppl Fig 6.

Corrected.

line 195: remove alternatively...

This phrase has been rephrased and Figure 3d was improved to show the alternation of seismicity pulses in the shallow and deep cluster.

line 196: seismicity rate anomalies, refer to a figure and indicate it. I can see seismicity rate in Fig 2e and 3d, but there are several peaks. Indicate time between responses.

Major seismicity rate anomalies have been remarked on Figure 3d, and the figure is now cited in the text. Time between these bursts cannot be robustly estimated, thus we prefer not to mention it.

line 196: You mention several times the lava emissions and seem to have a non-trivial role in the conceptual model. I request to include lava emissions in Fig 3 if you can get the data.

We do not have the full data to plot the timeline of the lava emissions. Cited references confirm the variability of the rates, but do not provide reliable time series.

line 199: "by late November"

Corrected.

line 208: "Our results". Your results cannot provide this evidence. Maybe petrological/geochemical, but if you don't have seismicity (silent), you cannot claim that.

The sentence has been rephrased to clarify “Our results of pre-eruptive seismicity location suggest the shallow reservoir was established early, although it was silent and undetected before the eruption. It’s location, hypothesised by petrological and deformation studies, corresponds to the starting point of the dikes that reached the surface (Fig. 2a,b)”

line 214: Fig 4d, are you referring to the dark green beachball? mention the color you are referring.
Corrected.

line 226: lava emission
line 229: lava emission
Corrected.

line 232: citation for magma pulses producing doublets
We suggested a possible interpretation for earthquake doublets in this environment, attributing them to the ascent of small magma batches. This could either (1) trigger multiple earthquakes in short time over neighboring seismogenic structures or (2) generate reversed temporal stress perturbation (i.e. first pressurizing, then depressurizing a conduit) over short time and thus trigger doublets on the same fault.
We are not aware of a similar observation which could be cited, but rephrased the text to make it clearer.

line 234: shallow cluster seismicity in two regions. I saw 3 colours in Fig 4 a-b. Refer to colours if you are grouping them.
This figure has been improved, now figure 4a,b show the normalized density of relocalized earthquakes and the two regions are clearly identified.

line 235: pressure axis
Corrected.

line 237-240: repetition of ideas. Also, mention the shear stress pattern change that enable the curved structure observed. The deep cluster seems to have the rotation in the opposite direction, when compared with the Suppl Fig 10.
We rephrased the text.

line 247: "curved path" and relationship with Fig 10a. The locations follow the enhanced shear stress areas for the case of N-S background stress. Fig 4c and 5c show NNW-SSE orientation of the background stress. How this will affect the results? Is it within the errors?
We improved the manuscript (see reply to the last comment below)

line 255: why monogenetic only?
We have removed the word monogenetic from that sentence. All the observations obtained from this study can help to understand the behavior of the volcano and improve early warning and monitoring.

Methods

line 279: Velocity structure: How does the velocity structure affect the shallow locations? How the residuals behave?
I can see more dispersion in shallow seismicity (depths ~3km). Can it be due to constant properties in quite a thick layer (0-5 km)?
Shallow <5 km seismicity is not discussed in the article. Shallow seismicity provides critical insight for monitoring purposes when the crisis is imminent.
We agree that shallow seismicity shows a higher dispersion and larger residuals in IGN catalog and relocalization. This is probably produced by two factors: the use of a simplified velocity model in the shallower part of the crust and the difficulty of phase picking for such seismicity. This seismicity is not critical for our study, as we are focusing on the deeper seismicity and the magma

plumbing system below the island. A distributed shallow pre-eruptive seismicity may be interpreted by broader fracturing processes at the surface when the dyke reaches a very shallow depth.

Moment tensor inversion and stress partitioning

line 286: *mbLg* was not mentioned in the main body of the article. Also, I couldn't see events with magnitude larger than 4 in Fig 3a. Which deep events were used? Is the plot 3a missing larger events?

We simplified the text, referring only to ML which is the scale used in plot 3a. No larger events are missing.

mbLg was the local magnitude used by IGN in its reports during the eruption. The recent work of Rueda and Mezcua 2022 showed that *mbLg* lead to serious distortions in volcanic areas and derived a specific scale, ML, for the Canarian Archipelago. Bulletins of the eruption can now be downloaded with both magnitude scales from (<https://www.ign.es/web/ign/portal/vlc-catalogo>).

line 299: not local, regional.

Corrected

Seismicity classification

I suggest to use one classification. It is confusing to use 3 different criteria.

We decided to show only MT clustering results in the main figures of the article and move the other 2 classifications (spatial and waveform similarity clustering) to the Supplementary Material (see reply to main comment #1).

Figure 1:

Larger fonts in the inset

Corrected.

Figure 3

Keep the Y-labels aligned.

No $ML > 4$ for MT inversion?

Figure 3 plots magnitude ML, only one event with $ML > 4.0$ (19/11/2021 01:08 (UTC)).

A new figure with the $M_w(ML)$ relation has been added to Supplementary Material (Suppl. Fig 8)

3d - indicate the relationship between deep and shallow bursts with arrows.

Use stages or phases, but no both. I suggest stages to avoid confusion with P and S phases.

Corrected.

Figure 4:

light green beachball in plan view and profile is not consistent.

Shallow cluster was shown in purple-ish in Figs 1,2 and 3. Keep consistent.

Use one clustering criteria, this Figure is extremely confusing.

line 420: "identified as in a". This Figure has 4 beachballs and 3 colours in a. Not consistent.

line 421: pressure axes

Corrected.

Figure 5:

Deep cluster was shown in green-ish in Figs 1,2 and 3. Keep consistent.

Corrected.

Figure 6:

indicate the curved dyke propagation with arrows.

This figure shows a vertical projection by longitude where the aspect ratio is conserved (same projection as in Figure 1b). We cannot show the curvature of the dyke in this projection (the curvature is visible in the projection of Figure 1a). Seismicity associated with the dyke is coloured in red in Fig 1b and in the scheme we represented it by straight lines and an arrow at the end.

Supplementary material:

Fig 4: Example of inversion of a 3.6 event, in the text says inversions are carried out for events with magnitudes above 3.8 for shallow cluster

Text referred to ML magnitude instead of mbLg.

Fig 8: Why not show the stacked waveforms?

Stacked waveforms have been added in black color at the bottom of each family (now Suppl. Fig 11).

Fig 9: I can see a more important normal/thrust behaviour in the deep cluster, contrary to what the text says.

We think the reviewer refers here to the different types of mechanism in the deep cluster compared to the shallow cluster. In the deep cluster, many focal mechanisms are still strike-slip to oblique, but others actually show important thrust or normal components. We agree and we have added one sentence to state this.

line 89: Definition of R equal to 1.

Corrected.

Fig 10: the locations fit with the numerical simulation when the pressure axis is N-S, which is not the real case. There must be a rotation ~30 degrees

The manuscript text here was not clear enough and we have improved it. A previous study (Geyer et al 2016) suggested the regional stress to be oriented ~N25°E. While this may be an acceptable estimation deeper in the mantle and at a regional scale, it does not seem very accurate in the crust beneath La Palma, where we believe the orientation of σ_1 is rather NS (and σ_3 EW), possibly also affected by the island topography. A strong argument here is given by the NS trend of the dike in its final path away from the reservoir, and its EW opening (σ_3). Even the ~NS orientation progression of former volcanism at La Palma and the island topography confirm this stress orientation.

Rodrigo Contreras-Arratia

REVIEWERS' COMMENTS

Reviewer #1 (Remarks to the Author):

I have read and reviewed the revised version of "Magmatic plumbing and dynamic evolution of the 2021 La Palma (Spain) eruption" by del Fresno et al. I am happy with the changes the authors have made to the manuscript and figures and have only some minor wording suggestions as well as two regarding the structure of the text as outlined below that would further improve the readability of the manuscript.

Line 105: Grammar. Move 'throughout the text' towards the end of the sentence

Line 109: something is missing in this sentence, maybe a "where" after the first comma?

Line 111: Wording. Either "peaks showing intensification" or "seismicity rates showing intensification" etc

Line 115: Wording. "The activity in the deep cluster"

Line 118: Two points

Line 160: This paragraph is in the wrong place as you already refer to the clusters earlier. Please move this up in the main text.

Line 170ff: For the reader, it may be easier to follow your explanations if you clearly state from the beginning of the discussion that Figure 6 is a conceptual model which you use to explain your findings. Your refer to the figure but it is not clear in what context and then you only mention it explicitly in the very last paragraph of the discussion section. I suggest you state in the first paragraph of the you have built a conceptual model which you will describe using your observations. Then it's much easier for the reader to follow and you can keep referencing it for the mechanisms you describe.

Line 188: maybe I missed it, but I think you need to explain σ_1 and σ_3 to your audience.

REVIEWERS' COMMENTS

In the following paragraphs, we reply to the reviewer #1 comments. We mark in black the comments by reviewer #1, and in blue our reply.

Reviewer #1 (Remarks to the Author):

I have read and reviewed the revised version of “Magmatic plumbing and dynamic evolution of the 2021 La Palma (Spain) eruption” by del Fresno et al. I am happy with the changes the authors have made to the manuscript and figures and have only some minor wording suggestions as well as two regarding the structure of the text as outlined below that would further improve the readability of the manuscript.

We are thankful for the review and we accounted for all remaining suggestions.

Line 105: Grammar. Move ‘throughout the text’ towards the end of the sentence

We decided to better remove “throughout the text” from this sentence.

Line 109: something is missing in this sentence, maybe a “where” after the first comma?

Corrected.

Line 111: Wording. Either “peaks showing intensification” or “seismicity rates showing intensification” etc

Corrected with “seismicity rates showing intensification”.

Line 115: Wording. “The activity in the deep cluster”

Line 118: Two points

Both corrected.

Line 160: This paragraph is in the wrong place as you already refer to the clusters earlier. Please move this up in the main text.

We moved this paragraph up in the main text.

Line 170ff: For the reader, it may be easier to follow your explanations if you clearly state from the beginning of the discussion that Figure 6 is a conceptual model which you use to explain your findings. You refer to the figure but it is not clear in what context and then you only mention it explicitly in the very last paragraph of the discussion section. I suggest you state in the first paragraph of the you have built a conceptual model which you will describe using your observations. Then it’s much easier for the reader to follow and you can keep referencing it for the mechanisms you describe.

Done.

Line 188: maybe I missed it, but I think you need to explain σ_1 and σ_3 to your audience.

We improved the sentence.